# Reasons for the High Electrical Conductivity of Bismuth Ferrite and Ways to Minimize It

Konstantin P. Andryushin [1,*], Vladimir P. Sakhnenko [1], Anatoliy V. Turik [1,†], Lidiya A. Shilkina [1], Alexey A. Pavelko [1], Svetlana I. Dudkina [1], Angela G. Rudskaya [2], Daniil D. Rudskiy [1], Iliya A. Verbenko [1], Sidek V. Hasbulatov [3], Larisa A. Reznichenko [1], Ivan A. Parinov [4], Shun-Hsyung Chang [5] and Hung-Yu Wang [5]

[1] Research Institute of Physics, Southern Federal University, Rostov-on-Don 344090, Russia; vpsakhnenko@sfedu.ru (V.P.S.); lid-shilkina@yandex.ru (L.A.S.); dipoleglass@gmail.com (A.A.P.); s.i.dudkina@yandex.ru (S.I.D.); rudskiy@sfedu.ru (D.D.R.); ilich001@yandex.ru (I.A.V.); lareznichenko@sfedu.ru (L.A.R.)

[2] Faculty of Physics, Southern Federal University, Rostov-on-Don 344090, Russia; agrudskaya@sfedu.ru

[3] Department of Physical electronics, Chechen State University, pr. Dudaev Boulevard, 17a, Grozny 366007, Russia; said_vahaevich@mail.ru

[4] I. I. Vorovich Mathematics, Mechanics and Computer Sciences Institute, Southern Federal University, Rostov-on-Don 344090, Russia; parinov_ia@mail.ru

[5] Department of Microelectronics Engineering, National Kaohsiung University of Science and Technology, Kaohsiung 807, Taiwan; stephenshchang@me.com (S.-H.C.); hywang@kuas.edu.tw (H.-Y.W.)

[*] Correspondence: kpandryushin@gmail.com; Tel.: +8-8632-43-40-66

[†] The author has passed away.

**Abstract:** The reasons for the high electrical conductivity of bismuth ferrite due to its natural composite structure, structural non-stoichiometry, redox processes, and the boundary position in the perovskite family have been considered. It has been shown that it is possible to significantly (2–3 orders of magnitude) reduce the conductivity of $BiFeO_3$ by introducing large-sized ions of rare-earth elements (REE: La, Pr, Nd, Sm, Eu, Gd with $0.94 \leq \overline{R} \leq 1.04$ Å) in amounts of up to 10 mol %. An interpretation of the observed effects has been given. A consideration about the appropriateness of taking into account the presented results when developing devices using materials of the $BiFeO_3$/REE type has been expressed.

**Keywords:** ceramic; $BiFeO_3$; rare-earth elements; electrical conductivity; Maskwell-Wagner relaxation

## 1. Introduction

Features of modern technology: intensification of processes associated with an increase in operating temperatures, pressures, frequencies; acceleration of energy transformations; and the pursuit of multifunctionality of high-tech products all determine tougher technical, economic and environmental requirements demanded for the material and technological base used in industry. In this regard, multiferroic media with coexisting special electrical and magnetic properties are of increasing scientific and practical interest. These objects are promising for applications in spintronics, as well as in areas related to artificial intelligence with other high-tech areas of the real sector of world economies [1,2]. At the same time, the most attractive are compositions containing transition metal ions, which cause magnetic phase transitions and ions with an unshared pair of *s* electrons capable of exhibiting a ferroelectric state, such as Bi, Pb, one of such compounds is $BiFeO_3$ [3]. $BiFeO_3$ has a relatively simple structure (rhombohedrally distorted perovskite, space group–*R3c*), high temperatures of ferroelectric ($T_c$ = 1083 K) and antiferromagnetic ($T_N$ = 643 K) transitions, which, in many respects, define it as the most convenient object of experimental research [4–6]. However, a number of features of this compound, high electrical conductivity and the presence of a spin modulation cell that is not commensurate with the period of the unit

cell, make it extremely difficult to perform a comprehensive study of this material or put it into practice in the areas most in demand.

In a number of works it has been shown experimentally [7,8] and theoretically [9] that under the conditions of solid-phase synthesis and sintering using conventional ceramic technology (without externally applied pressure) $BiFeO_3$ is not thermodynamically stable. The resulting product inevitably contains one or another amount of impurities, which, of course, leads to a decrease in its electrical resistance. At the same time, a low level of reproducibility and inconsistency of the results obtained by various groups of researchers becomes another important problem, which is due to the critical dependence of the target product not only on the conditions of synthesis and sintering, but also on the whole of its thermodynamic history.

Based on a series of experiments, the features of $BiFeO_3$ phase formation have been studied in detail [10]. It has been shown that a significant role is played by processes caused by the structural non-stoichiometry of the objects, the appearance of which, due to both the high stability of the intermediate phases and the loss of oxygen as a result of partial reduction of the elements with a variable oxidation state, can also influence the $BiFeO_3$ macroresponses and its electrical conductivity. Based on the results obtained we proposed the methods for optimizing the technology of obtaining materials involving $BiFeO_3$.

In this paper, we examine in greater detail the reasons for the increased conductivity of $BiFeO_3$ and the possibility of minimizing it by variations in the cationic composition upon modification of this multiferroic with rare-earth elements (REE).

## 2. Materials and Methods

The objects of study were solid solutions (SS) of the $Bi_{1-x}REE_xFeO_3$ composition (REE, lanthanides, Ln: La, Pr, Nd, Sm, Eu, Gd, Tb, Dy, Ho, Er, Tu, Yb, Lu; $x = 0.0–0.5$; $\Delta x = 0.025$, 0.05, 0.10).

The main method for preparing studied SS was solid-phase synthesis with intermediate grinding and granulation of dispersed crystalline powders and subsequent sintering using conventional ceramic technology (without externally applied pressure). Optimum technological procedures ensuring the purity (or the minimum number of ballast phases), experimental density close to the theoretical one, and the integrity of ceramics were chosen on a series of the samples of each composition, varying temperature, duration and frequency of firing. Specific modes of the synthesis and sintering of the SS were: $T_1 = (1060 \div 1180)$ K, $\tau_1 = (5 \div 10)$ hours; $T_2 = (1070 \div 1180)$ K, $\tau_2 = (5 \div 10)$ hours, $T_{sint} = (1140 \div 1270)$ K, $\tau_{sint} = 2$ h (depending on the composition). Technological regimes for the preparing of studied SS are presented in [11].

Some SS were subjected to mechanically activating procedures performed with the help of a planetary mill of the AGO-2 brand. Before sintering the samples, the powders were formed in the shape of columns with a diameter of 12 mm and a height of 20 mm. Sintered ceramic billets were machined (plane cutting, grinding on flat surfaces and ends) in order to obtain measuring samples with a diameter of 10 mm and a height of 1 mm. Each composition of such samples contained $(5 \div 10)$ pieces. Before metallization, the samples were calcined at a temperature of $T_{calc} = 770$ K for 0.5 h to remove residual organic substances and degrease surfaces in order to increase the adhesion of the metal coating with ceramics. The electrodes were applied by double burning of a silver-containing paste at a temperature of 1070 K for 0.5 h.

X-ray diffraction studies at room temperature were performed by powder diffraction using DRON-3 and ADP-1 diffractometers (Соκα–radiation; Mn–filter; Bragg-Brentano focusing scheme). We studied bulk and ground ceramic objects, which allowed us to exclude the influence of the surface effects, strength and textures that arise in the process of obtaining ceramics. The calculation of the structural parameters was performed according to a standard method. X-ray density ($\varrho_{X-ray}$) was determined by the formula: $\varrho_{X-ray} = 1.66 \cdot M/V$, where $M$ is the weight of the formula unit in grams, $V$ is the volume of the perovskite cell in Å. The measurement errors of the structural parameters were as fol-

lows: linear $\Delta a = \Delta b = \Delta c = \pm(0.002 \div 0.004)$ Å; angular $\Delta \alpha = 3'$; and volume $\Delta V = \pm 0.05 \text{Å}^3$ ($\Delta V / V \cdot 100\% = 0.07\%$).

The experimental ($\varrho_{\text{exp}}$) density of the samples was determined by hydrostatic weighing, where octane was used as a liquid medium. The relative density ($\varrho_{\text{rel}}$) was calculated by the formula: ($\varrho_{\text{exp}} / \varrho_{\text{X-ray}}$)$\cdot 100\%$. Densities of the studied SS are presented in [12].

The real and imaginary parts of the relative permittivity and dielectric loss tangent ($\varepsilon' / \varepsilon_0$, $\varepsilon'' / \varepsilon_0$, tg$\delta$) at frequencies of ($20 - 2 \cdot 10^6$) were studied on a special bench using a precision LCR- meter of Agilent E4980A in the intervals of (300–900) K temperatures.

The electrical resistance R (Ohm) was measured at direct current at room temperature using an Agilent 4339B High Resistance Meter. The electrical resistivity, $\varrho_{\text{v}}$, was calculated by the formula: $\varrho_{\text{v}} = R \cdot S / l = R \cdot \pi d^2 / 4l$, and the electrical conductivity, $\sigma$, was calculated by the formula: $\sigma = 1 / \varrho_{\text{v}}$, where l is the sample thickness (m); $S$ is the flat surface area of the sample (m$^2$); $d$ is the diameter of the sample (m).

## 3. Results

### 3.1. About the Causes of the High Conductivity of BiFeO$_3$

Figure 1 shows the temperature dependences of $\varepsilon' / \varepsilon_0$, $\varepsilon'' / \varepsilon_0$ BiFeO$_3$ and BiFeO$_3$/REE on different frequencies of the *ac* field.

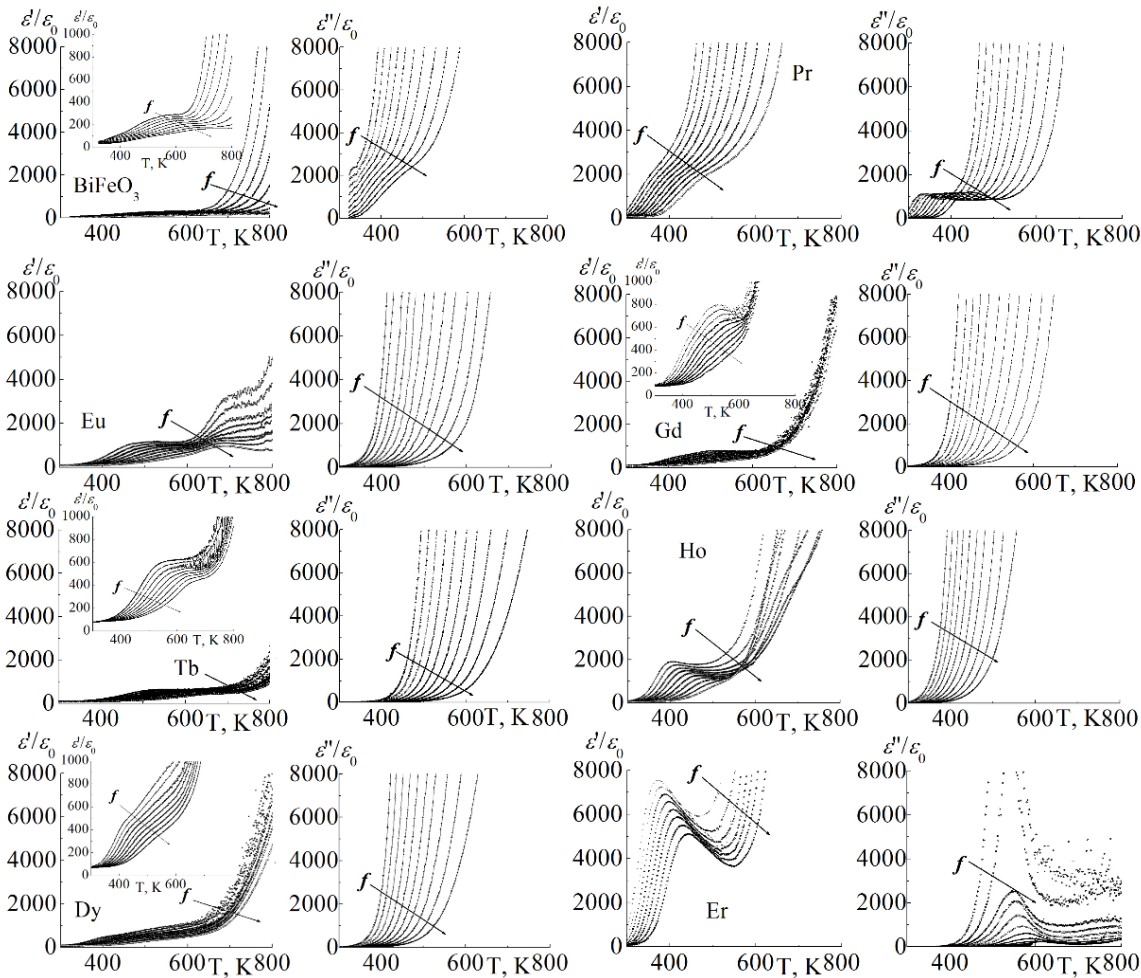

**Figure 1.** Temperature dependences of $\varepsilon' / \varepsilon_0$, $\varepsilon'' / \varepsilon_0$ BiFeO$_3$ and BiFeO$_3$/REEx ($x$ = 0.1) in the frequency range of ($25 \div 2 \cdot 10^6$) Hz of the ac field.

The following facts are noteworthy: the enormous values of these quantities in the interval of ($400 \div 800$) K (1); the formation of a low-temperature relaxation maximum $\varepsilon' / \varepsilon_0$ at ($400 \div 500$) K, not associated with any of the known phase transitions (2); rapid growth

of $\varepsilon'/\varepsilon_0$ at 600 K at all frequencies; the form in the range of $(600 \div 700)$ K of a "dome" of this characteristic (3); the beginning of the formation of a "dome" $\varepsilon''/\varepsilon_0$ at high frequencies at 800 K(4).

To describe the process of low-temperature dielectric relaxation according to the data obtained from the dependences $\varepsilon'/\varepsilon_0$ (T) and $\tan\delta$ (T), the graphs $\ln\omega$ $(1/T_m)$ ($T_m$ are the temperatures of the extrema $\varepsilon'/\varepsilon_0$ and $\tan\delta$ measured at the frequency $f = \omega/2\pi$) presented in Figure 2.

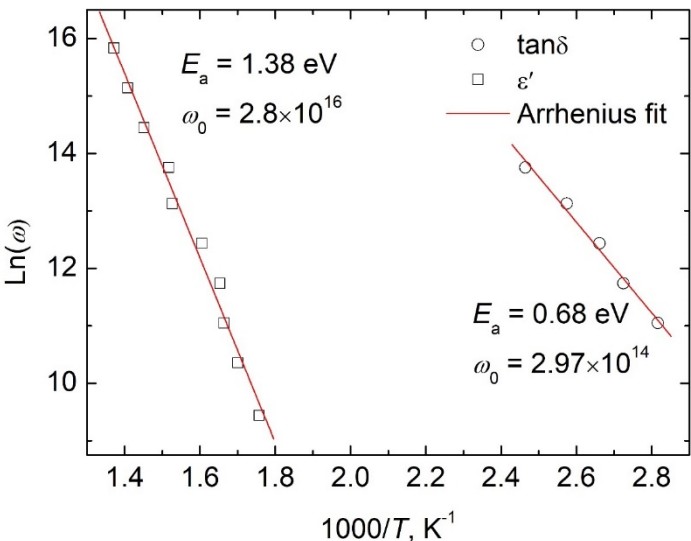

**Figure 2.** Arrhenius plots describing relaxation processes on $\varepsilon'/\varepsilon_0$ (T) and $\tan\delta$ (T) dependences of BiFeO$_3$ ceramics.

The presented dependences satisfy the Arrhenius law, which allows one to characterize the relaxation process using the activation energy, $E_a$, and the average frequency of overcoming the potential barrier, $\omega_0$. For the two plotted dependences, these parameters take significantly different values ($E_a$ = 1.38 and 0.68 eV, $\omega_0 = 2.8 \times 10^{16}$ and $2.97 \times 10^{14}$ rad/s, respectively), indicating two independent relaxation processes occurring in the object. Based on the literature data, it can be argued that such values are typical for the Maskwell-Wagner relaxation [13–15], associated with the accumulation of free charges on the interface of components in spatially inhomogeneous media on the background of interlayer, interphase, and intraphase structural rearrangements. The reason for the development of this situation, as well as the appearance of two relaxation processes, is the natural composite structure of BiFeO$_3$, which is formed on the basis of at least two (not counting BiFeO$_3$ itself) Bi-, Fe-containing compounds (Bi$_{25}$FeO$_{39}$, Bi$_2$Fe$_4$O$_9$) that almost always accompany the formation of BiFeO$_3$, remain in it (in different amounts) in the form of ballast phases [7,16–25] (we will talk about this in more detail below).

A brief mention of the physical mechanisms of macroevents developing in such heterogeneous media should be made. It is known that in such materials an increase in the permittivity caused by the Maxwell-Wagner polarization is detected [23], and it is associated with the accumulation of charges at the boundaries of inhomogeneities in dielectrics [26]. Moreover, the increase in dielectric permittivity is always due to the electrical conductivity of the components (crystallites) of a heterogeneous dielectric [27,28]. In experimental works [29–31], a gigantic increase in the permittivity is associated with relaxation polarization processes. In works [31,32] an analysis of the growth of the low-frequency permittivity of a heterogeneous dielectric, which is a randomly inhomogeneous object, is made using finite elements and Monte Carlo methods. The author shows that the greatest increase in the dielectric permittivity occurs in the so-called "reciprocal composite", that is, in two-layer structures in which the dielectric permittivity of one component is significantly greater than that of the second one, whereas the specific conductivity of the

second component is much higher than that of the first one. A number of works [33,34] develop the ideas presented in [31]. Meanwhile, the physical nature of the gigantic increase in the effective dielectric permittivity is understandable: it is connected, as mentioned above, with the charge accumulation at the interfaces of the components of a heterogeneous dielectric. At the initial moment, after applying an electric voltage to the sample, the electric induction in the layers is the same, which means the distribution of the field strength is inversely proportional to the dielectric permittivity of the layers. However, after completion of the processes, the densities of the conduction currents are equalized, and the field strength in the layers becomes inversely proportional to the specific conductivities of the layers, which leads to an increase in the charge accumulated at the boundaries precisely in the "reciprocal composite". The same physical mechanisms lead to an increase in effective electrical conductivity.

In the work of A.V. Turik and co-workers [35] it is quite rightly pointed out that in the works [30,36] and even in the most detailed work [31] devoted to the giant growth of $\varepsilon$, the possibility of achieving giant electrical conductivities is not even mentioned. It should be noted that in the work [37] an analysis of this possibility is performed, but only for some types of heterogeneous dielectrics. A complete analysis of this problem in the literature is not available, which has served as an incentive to focus on this issue in greater detail.

To predict the necessary conditions for giant dielectric amplification, it is sufficient to use the simplest model of an electrically inhomogeneous dielectric considered by Hippel [28]. Using the methods of impedance spectroscopy, for the electrical conductivity of a dielectric sample consisting of two parallel layers with significantly different capacitances and resistances, in *ac* fields with a circular frequency $\omega$ [35], we obtained the following formula:

$$G = [R_1 + R_2 + \omega^2\left(\tau_1^2 R_2 + \tau_2^2 R_1\right)]/[(R_1 + R_2)^2 + \omega^2(\tau_2 R_1 + \tau_1 R_2)]^2, \qquad (1)$$

and for capacity

$$C = [\tau_1 R_1 + \tau_2 R_2 + \omega^2(\tau_1 \tau_2(\tau_2 R_1 + \tau_1 R_2))]/[(R_1 + R_2)^2 + \omega^2(\tau_2 R_1 + \tau_1 R_2)]^2, \qquad (2)$$

where $\tau_1 = R_1 C_1$; $\tau_2 = R_2 C_2$. Here $R_1$ and $C_1$—resistance and capacitance of the first layer, $R_2$ and $C_2$—resistance and capacitance of the second layer.

From (1) and (2) for static values of $G$ and $C$ at $\omega \to 0$ it can be written:

$$G_s = 1/(R_1 + R_2), \quad C_s = [\tau_1 R_1 + \tau_2 R_2]/[\left(R_1 + R_2\right)^2 + R_1 + \tau_1 R_2)]^2, \qquad (3)$$

and for high-frequency values of G and C at $\omega \to \infty$ it can be written

$$G_\infty = \left(\tau_1^2 R_2 + \tau_2^2 R_1\right)/(\tau_2 R_1 + \tau_1 R_2)^2, \quad C_\infty = \tau_1 \tau_2/(\tau_2 R_1 + \tau_1 R_2)^2, \qquad (4)$$

Thus, the two-layer Maxwell-Wagner capacitor is a dielectric with monotonically decreasing real parts of the capacitance with an increase in frequency and the complex permittivity. In this case, the imaginary part of the dielectric permittivity has a relaxation maximum [31,35].

One of the mechanisms of the appearance of enormous electrical conductivity in $BiFeO_3$ is the peculiar structure of $BiFeO_3$ conditioned by the features of the state diagram of the $Bi_2O_3$–$Fe_2O_3$ binary system (Figure 3) with three bismuth ferrites of different quantitative elemental composition [31,38]. $BiFeO_3$ begins to form in the temperature range of $770 \leq T_1 \leq 820$ K; $Bi_{25}FeO_{39}$ (sillenite phase) appears earlier at lower temperatures of ~720 K and is an intermediate compound; $Bi_2Fe_4O_9$ (mullite phase) is present in $BiFeO_3$ at the stage of improving its structure ($920 \div 1020$ K) as a concomitant compound. Its formation is the result of structural A- non-stoichiometry of $BiFeO_3$ [39], a phenomenon typical of oxygen-octahedral compounds with the $ReO_3$ structure [40] and

perovskite [41–43] containing ions of variable valence. A specific feature of such substances is the ability of a crystallographic shift to exclude an appearance of anionic vacancies (for various reasons), and in complex oxides (of the perovskite type), their equal number of cuboctahedral positions. Thus, a decrease in the A-O positions is nothing more than a self-organizing compensatory mechanism for eliminating point defects (vacancies) due to the organization of extended defects–crystallographic shift planes, while maintaining the highly ordered structure of anion-deficient oxides. In this case, the excess A-component, depending on its size factor and thermodynamic background, can be placed in regular B-positions; at irregular locations, forming an autoisomorphic substance [44] or an internal multiplicative [45] SS of implementation; in intergranular layers; or precipitate in the form of impurities [46].

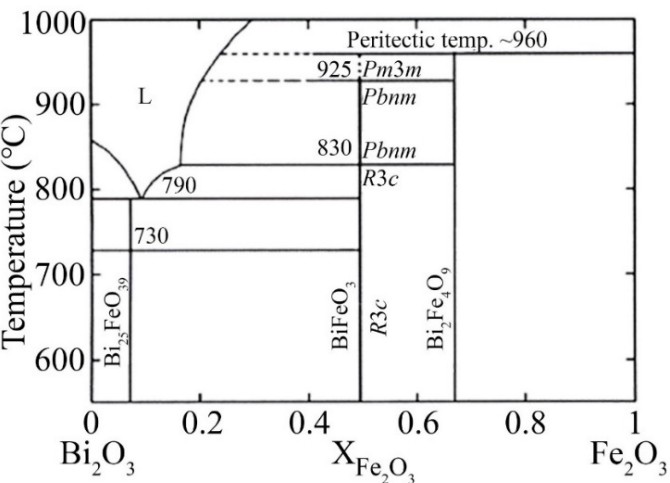

**Figure 3.** Phase diagram of the $Bi_2O_3$-$Fe_2O_3$ binary system.

　　　In our case, probably as a result of the deficiency of A-positions (according to our calculations, it is at least 3%), excessive $Bi^{3+}$ ions appear, which displace $Fe^{3+}$ ions from oxygen octahedra (as follows from [47], the placement of $Bi^{3+}$ in A- and B-positions is possible), thereby creating the prerequisites for the formation of the $Bi_2Fe_4O_9$ compound (with a higher Fe content). In $BiFeO_3$, the appearance of the above described vacancies in the A- and O-sublattices may be due to the partial sublimation of volatile $Bi_2O_3$ in the synthesis process of the compound [48] (according to the scheme $Bi^{3+}_{1-x}\square_x Fe^{3+}O^{2-}_{3-1.5x}\square_{1.5x}$, $\square$ is a vacancy), partial reduction of $Fe^{3+} \rightarrow Fe^{2+}$ (according to the scheme $Bi^{3+}Fe^{3+}_{1-x}Fe^{2+}_x O^{2-}_{3-x/2}\square_{x/2}$), the traditional loss of the part of oxygen in oxygen-octahedral media.

　　　Another mechanism for increasing σ of $BiFeO_3$ is based on structural instability due to its boundary position in a family of oxides with a perovskite type structure (OSP) (Figure 4, where $\mu$ is the degree of stretching of A-O bonds and compression of B-O bonds due to the discrepancy between the sizes of ions and the sizes of voids of the closest packing [47,49,50]), as well as the proximity of sintering and incongruent (with decomposition) melting temperatures [38,51]. This can lead to a slight violation of the stoichiometry of $BiFeO_3$ due to the formation of a noticeable amount of difficult to remove ballast phases [19,20], including unreacted initial components [52]. This is facilitated by the narrow concentration-thermal interval of the existence of the $BiFeO_3$ [8,9] phase, and quite wide crystallization regions of $Bi_2Fe_4O_9$ and $Bi_{25}FeO_{39}$ compounds [7,16,51,53], as well as the complexity of the reaction in the equimolar mixture of $Bi_2O_3 + Fe_2O_3$ [17,18].

　　　In [9] calculations of the thermodynamic potentials for the reactions of $BiFeO_3$ formation, from which it follows that it is not the most stable of the possible compounds in the $Bi_2O_3$–$Fe_2O_3$ system, are provided. In [8] the results of a study of its temperature stability during being obtained by the "wet" low-temperature method are presented, and it is noted that at elevated temperatures (>970 K) this compound decomposes slowly. Attempts to obtain pure $BiFeO_3$-based material using exotic, more energy-consuming and labor-intensive

methods than the solid-phase reaction method in combination with conventional ceramic technology at the sintering stage, were also made in [19,20,54–59]. However, it was not possible to synthesize a pure, single-phase, thermodynamically stable product suitable for producing ceramics.

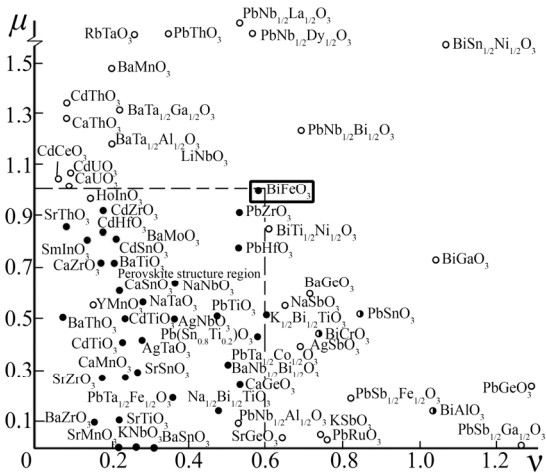

**Figure 4.** Diagram of directional parameters υ (characterizing the energy profitability of attaching atoms to each other along the directions of the highest concentration of electron density) and strength μ and the region of existence of oxides with a perovskite-type structure (OSP).

### 3.2. About Ways to Minimize the Conductivity of BiFeO₃

One of the ways to stabilize $BiFeO_3$ and optimize its properties is its introduction [60–63] into the REE composition. In this case, an increase in magnetoelectric coefficients is associated with the special magnetic properties of REE: despite the fact that their own ferromagnetic ordering occurs only at very low temperatures, the magnetic nature (*f*-ferromagnetism) of REE manifests itself in an increase in exchange interactions between other ferromagnetic ions, for instance, $Fe^{3+}$, which leads to an increase in the Néel temperature. This is facilitated by smaller (compared to Bi) radii of REE [44].

The introduction of sufficiently rigid highly ionized REE ions instead of easily deformable ions, for example, Bi, inevitably leads to a decrease in the Curie temperature and a convergence of the temperatures of the ferroelectric and antiferromagnetic transitions, which is highly desirable for practical applications. In addition, the substitution of part of $Bi^{3+}$ ions will lead to some increase in compositional disorder in the system, which will contribute to their manufacturability. The latter can be ensured by non-volatility and small-sized REE compared to $Bi^{3+}$. The thermal stability of $BiFeO_3$ modified with REE is increased, the range of optimal sintering temperatures widened, the phenomena associated with the melting of the synthesized product practically disappeared, the amount of impurities decreased. All these were prerequisites for the expected decrease in σ in $BiFeO_3$/REE.

Figure 5 shows the dependences of the electrical conductivity (at room temperature) on the concentration of REE introduced into $BiFeO_3$, and Figure 6 the same dependence on the radii of the modifying elements.

It can be seen that by doping $BiFeO_3$ with REE, in many cases it is possible to reduce σ and increase $\varrho_v$ by 2 ÷ 3 orders of magnitude. In this case, large-sized REE (tab.) with radii close to 0.99 Å, introduced in amounts of up to 10 mol %, turned out to be the most effective. This is explained, in our opinion, by the creation of the most favorable dimensional conditions for the existence of $BiFeO_3$ and, as a consequence, a decrease in impurity phases (which an increase in σ is associated with) and stabilization of the rhombohedral structure typical of $BiFeO_3$. Figure 7 shows the dependences of the structural characteristics of SS with large REE on the content (*x*) of the latter. It is clearly seen that in all the cases the state diagrams contain five concentration regions corresponding to

successive structural transformations $Rh + O_1 + O_2 \rightarrow O_1 + O_2 \rightarrow O_2$ with small variations in phase filling, and Figure 8 illustrates the dependences of the position of the boundaries of the existence of the pure $Rh$- phase, its mixture with the $O_1$- and $O_2$- phases, the values $(\Delta R_A/R_{min}) \cdot 100\%$ (where $\Delta R_A = R_{Bi} - R_{REE}$, $R_{min} = R_{REE}$), the differences of $(\Delta V)$ between $V_{exp}$ and $V_{theor}$ on the radius, $R$, REE. At lower $R$ (<0.99 Å), the shift of the $Rh$- boundary toward lower REE concentrations is obviously a consequence of a significant deviation of the conditions from those necessary for the formation of SS of substitution ($\Delta R_A/R_{min}$ should be equal, according to various estimates, to less than 15% [64] or 28 ± 2% [65]). The latter is associated with the loss of stability of the region of the $Rh$- and $O_{1,2}$- phases coexistence due to the greater instability of the structure of multiphase SS compared to single-phase ones.

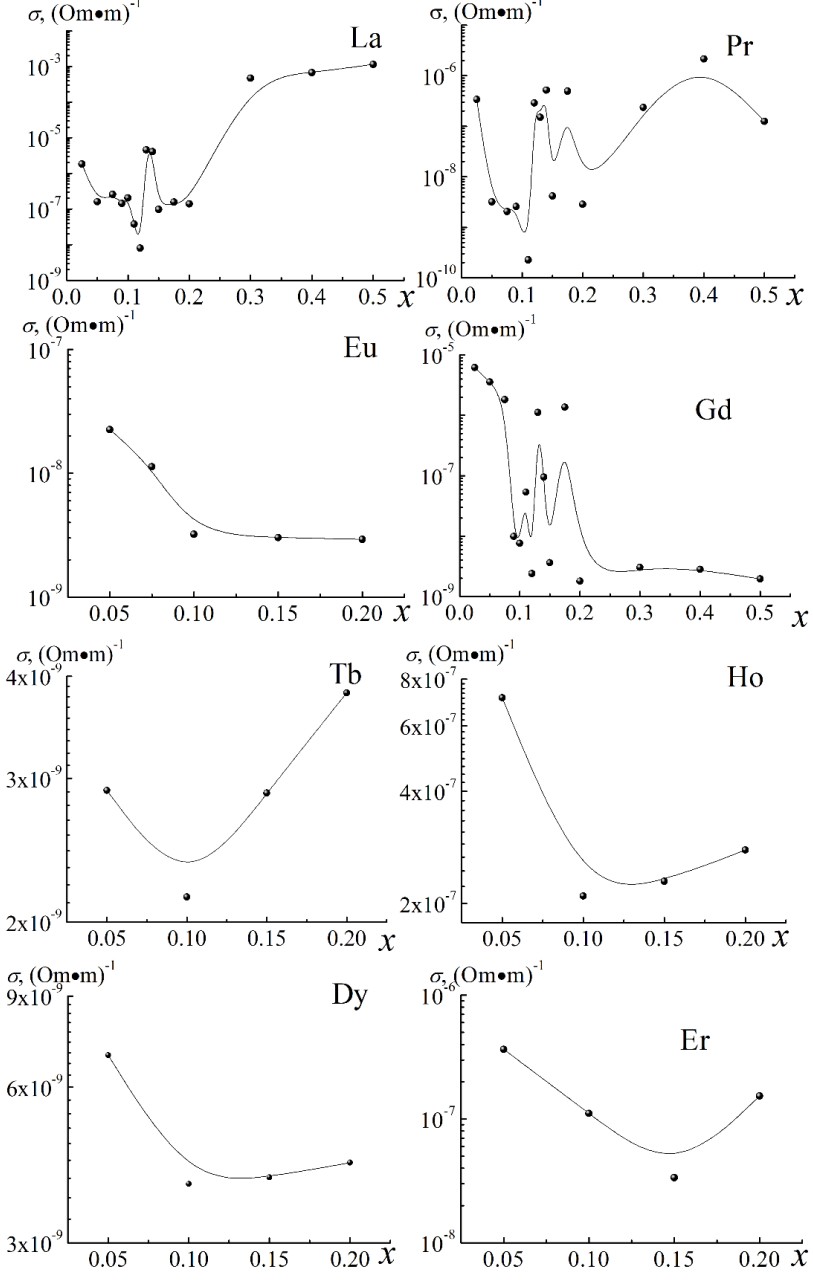

**Figure 5.** Dependences of the electrical conductivity $\sigma$ of BiFeO$_3$/REE–La, Pr, Eu, Gd, Tb, Ho, Dy, Er on the concentration of introduced $x$ of REE (experimental points are marked with markers; solid lines serve as a guide to the eye).

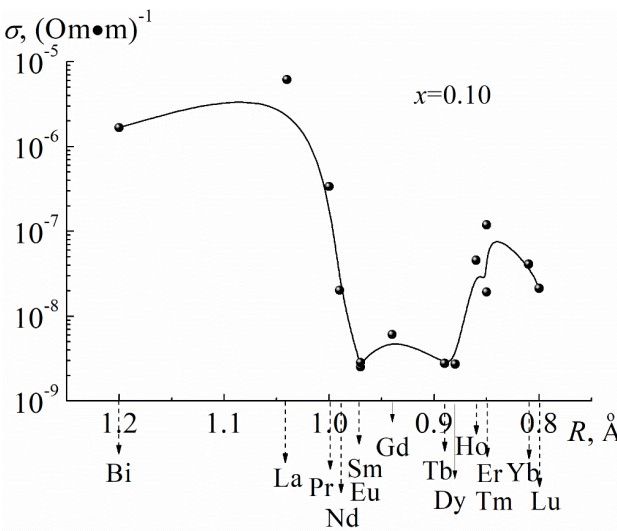

**Figure 6.** Dependences of the values of specific electrical conductivity $\sigma$, $BiFeO_3$/REE on radii, $R$, REE (experimental points are marked with markers; solid lines serve as a guide to the eye).

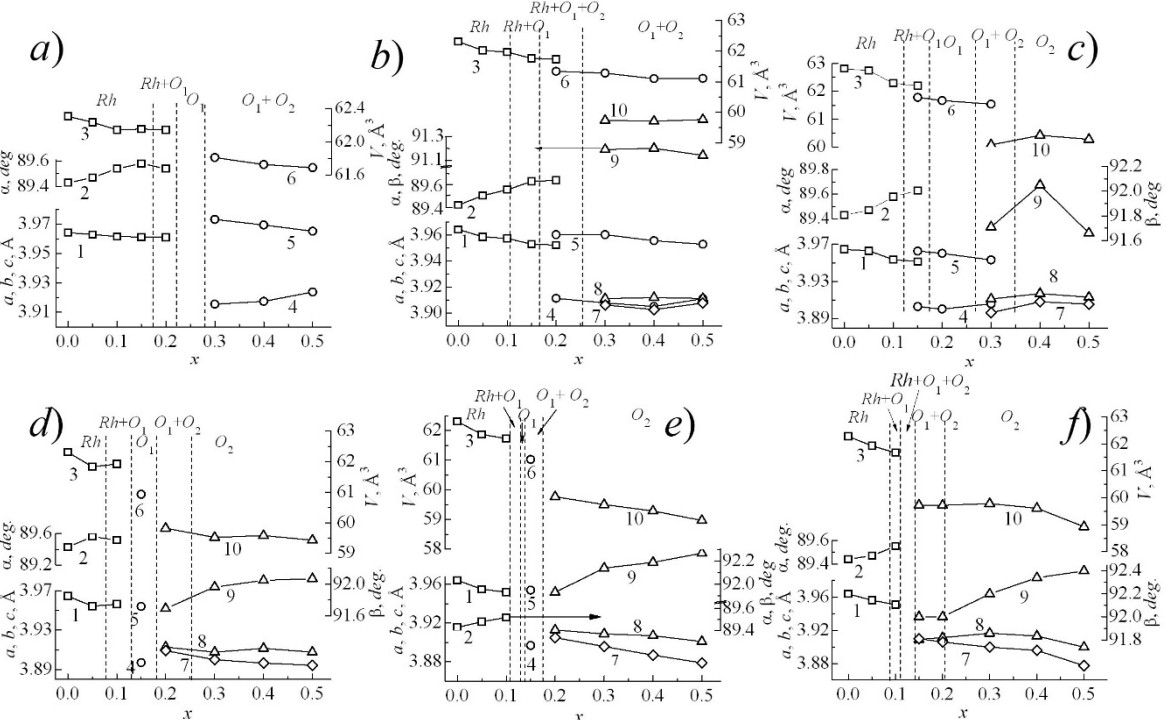

**Figure 7.** Dependences of the parameters of the perovskite cell of $BiFeO_3$ modified with La (**a**), Pr (**b**), Nd (**c**), Sm (**d**), Eu (**e**), Gd (**f**), respectively: 1, 2, 3–$a$, $\alpha$, $V$–parameters and volume of rhombohedral cell (*Rh* phase); 4, 5, 6–$c$, $a$, $V$–parameters and volume of the pseudotetragonal cell (orthorhombic phase $O_1$ of the $PbZrO_3$ type), 7, 8, 9, 10–$b$, $a$, $\beta$, $V$–parameters and volume of the monoclinic cell (orthorhombic phase $O_2$ of the type $GdFeO_3$).

The reasonableness of the made assumptions is also confirmed by the "behavior" of $\Delta V$ value (Figure 8b,d), which characterizes the structural nonstoichiometry of $BiFeO_3$. At low contents (~5 mol %) of REE, its effect is minimal, and $\Delta V_{0.05}$ on the background of a noticeable spread in $\Delta V$ values practically does not depend on $R$ of REE.

For larger values, in the region with $R > 0.99$Å $\Delta V$ changes little, contributing to the stabilization of the *Rh*-phase, and at $R < 0.99$ Å in the case with $x = 0.10$ $\Delta V_{0.10}$ logically increases with decreasing $R$, destabilizing *Rh*-, *Rh* +$O_{1,2}$- phases.

It should be also noted that our preliminary estimation calculations have shown that the introduction of the above mentioned REE into $BiFeO_3$ in small quantities leads to a decrease in the chemical bond parameter $\mu$, characterizing its strength, and in $BiFeO_3$ equal to the critical value for OSP ~ 1.04, which "shifts" $BiFeO_3$ into the region of OSP (Figure 4), preventing the possible loss of stability of its structure and favoring an increase in its electrical resistance.

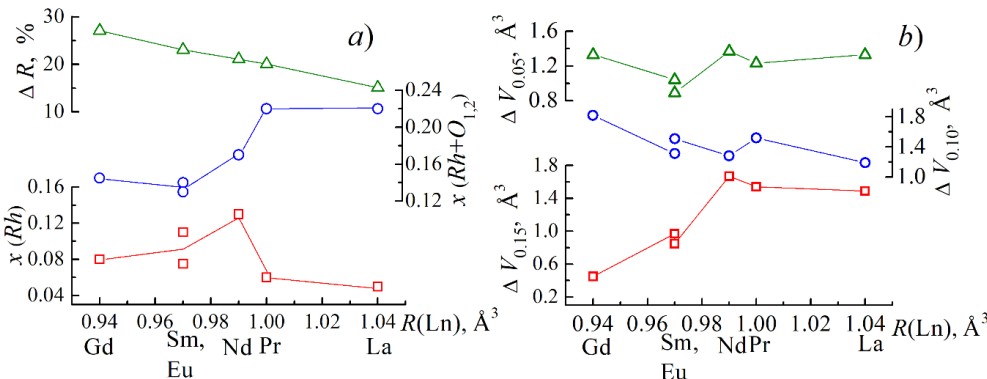

**Figure 8.** Dependences of the position of the boundary ($x$) of the existence of a pure *Rh*-phase, its mixture with $O_{1,2}$-phases, the relative difference between $R_{Bi}$ and $R_{REE}$ ($\Delta R$) on the radius, $R$, of REE (**a**); (**b**) Dependences of the difference between $V_{exp.}$ and $V_{theor.}$ ($\Delta V$) on the ionic radius, $R$, REE in SS with $x$ = 0.05, 0.10, 0.15.

This question should be considered in greater detail. During isovalent substitution, the parameters of the reduced perovskite cell change linearly according to SS concentrations. This is fixed by the so-called Vegard rule [66] and follows directly from the model of quasielastic cation–anion bonds in the cubic perovskite lattice [47,67,68]. Thus, we can write $a_{Bi,Ln}(x) = (1-x)a_{Bi} + xa_{Ln}$, where $a$ is the parameter of the given perovskite cell. It is convenient to associate the loss of stability of the perovskite structure with reaching the breaking point of the stretched (A-O) bonds and breaking them at some critical relative strain of $\Delta_{crit}/L_O$, where $\Delta_{crit}$ is the critical deformation of the bond, and $L_O$ is the length of the unstrained bond [47]. In the general case, the degree of the bond deformation is characterized by the values of $\Delta_{Bi} = a_{Bi,Ln}/\sqrt{2} - L_{A(Bi-O)}$; $\Delta_L = a_{Bi,Ln}/\sqrt{2} - L_{A(L-O)}$ and the corresponding relative quantities of $\Delta_{Bi}/L_{Bi-O}$ and $L_n/L_{Ln-O}$.

Following [47,67,68], the average parameter of the given perovskite cell, $a$, is $(n_A a_A + n_B a_B)/(n_A + n_B)$, where $n_{A,B}$ are the valencies of A-, -B- ions. At $n_A = n_B = 3$ (our case) $a = (a_A + a_B)/2$. Taking into account that $a_A = \sqrt{2}L_{A-O}$, $a_B = 2L_{B-O}$ [47], we obtained $a = (\sqrt{2}L_{A-O} + 2L_{B-O})/2$. In the Table 1, the $L_A$ values for Bi and REE, as well as $a_{BiFeO_3}$, $a_{LnFeO_3}$, and $\Delta_{Bi,Ln}$ and $\mu$ calculated using these data are presented. To calculate $\mu$, we use the formulas: $\mu_i = 7/2 \cdot \sqrt{n_s + \frac{9}{2}n_p^2 \cdot \Delta_{Bi}}$ for *Sp*-atoms and $\mu_i = (2 \cdot \sqrt{n_s} + 6n_d^3)\Delta_{Ln}/L_{(Ln-O)}$ for $ds^2$—valence electrons. In the first case, $n_S = 0$, $n_B = 3 \rightarrow \mu_i = 9/2 \cdot 9$; $0.07/2.78 = 1.04$~$1.0$ and $\mu = \Sigma \varepsilon_i \mu_i = 1$ (fractions of ions in A- positions), since in $BiFeO_3$ $\mu = \mu_i$. In the second case, $n_d = 1$, $n_S = 2 \rightarrow \mu_i = (2\sqrt{2} + 6 \cdot 1^3) \cdot 0.11/2.69 = 0.36$ (LaFeO$_3$). In SS $\mu = \sum \varepsilon_i \cdot \mu_i = (1 - x)\mu_{Bi} + x\mu_{Ln}$. Figure 9 shows the dependences of the calculated $\mu$ values in $Bi_{1-x}REE_xFeO_3$ SS with $x$ = 0.1; 0.2; 0.3 on the radii of Bi and REE. A sharp decrease in $\mu$ upon introduction of REE into the $BiFeO_3$ structure can be seen, while the minimum value of $\mu$ is realized in SS with La. Thus, the modification of $BiFeO_3$ with REE, contributing to a decrease in $\mu$, favors the stability of the structure, stabilization of the stoichiometric composition of $BiFeO_3$ and, as a consequence, a decrease in the electrical conductivity of the material.

**Table 1.** Main crystal chemical characteristics of A-elements (A-Bi, La, Pr, Nd, Sm, Eu, Gd, Ho).

| A-Element | $R$, Å | $\mu$, rel. Units | $L_{A\text{-}O}$, Å | $a$, Å | $\Delta_{A\text{-}O}$, Å | $\Delta_{A\text{-}O}/L_{A\text{-}O}$, rel. Units | Electronic Configurations of Outer Layers |
|---|---|---|---|---|---|---|---|
| Bi | 1.20 | 1.04 | 2.73 | 3.95 | 0.070 | 0.030 | $(S^2)p^3$ |
| La | 1.04 | 0.36 | 2.69 | 3.93 | 0.110 | 0.040 | $(f^0)ds^2$ |
| Pr | 1.00 | 0.39 | 0.65 | 3.90 | 0.116 | 0.044 | $(f^2)ds^2$ |
| Nd | 0.99 | 0.40 | 0.63 | 3.88 | 0.120 | 0.045 | $(f^3)ds^2$ |
| Sm | 0.97 | 0.44 | 2.62 | 3.88 | 0.130 | 0.050 | $(f^5)ds^2$ |
| Eu | 0.97 | 0.47 | 2.60 | 3.86 | 0.140 | 0.054 | $(f^6)ds^2$ |
| Gd | 0.94 | 0.50 | 2.59 | 3.86 | 0.148 | 0.057 | $(f^7)ds^2$ |
| Ho | 0.86 | 0.52 | 2.55 | 3.83 | 0.150 | 0.059 | $(f^{10})ds^2$ |

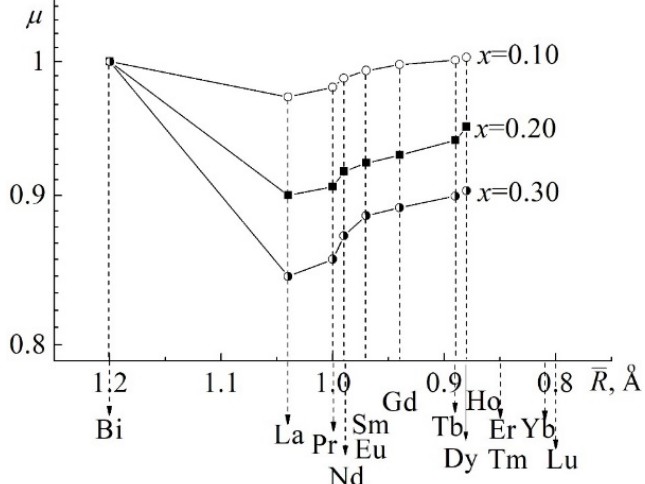

**Figure 9.** Dependences of $\mu_{REE}$ on radii, $R$, in $BiFeO_3$ SS with $x$ = 0.1; 0.2; 0.3.

## 4. Conclusions

Solid solutions of the composition $Bi_{1-x}REE_xFeO_3$ (rare-earth elements, lanthanides, Ln: La, Pr, Nd, Sm, Eu, Gd, Tb, Dy, Ho, Er, Tu, Yb, Lu; x = 0.0–0.5; $\Delta x$ = 0.025, 0.05, 0.10) were prepared by two-stage solid-phase synthesis followed by sintering by conventional ceramic technology with varying annealing temperatures. Comprehensive studies of the structure and specific electrical conductivity of the obtained experimental samples have been carried out.

The study of the temperature dependences of $\varepsilon'/\varepsilon_0$, $\varepsilon''/\varepsilon_0$ $BiFeO_3$ and $BiFeO_3$/REE on different frequencies of the *ac* field showed the presence of a low-temperature relaxation maximum $\varepsilon'/\varepsilon_0$ $(T)$ at $(400 \div 500)$ K, which is not associated with any of the known phase transitions in $BiFeO_3$ and satisfies the Arrhenius law with $E_a$ = 1.38 and 0.68 eV, $\omega_0$ = 2.8·10$^{16}$ and 2.97·10$^{14}$ rad/s, for $\varepsilon'/\varepsilon_0$ $(T)$ and tanδ (T), respectively. The results obtained allowed us to make conclusions about the presence of Maskwell-Wagner relaxation, as well as to consider the physical mechanisms of macroevents developing in such heterogeneous media.

The possibility of a significant (by 2–3 orders of magnitude) decrease in the electrical conductivity of $BiFeO_3$ by introducing into it large-size ions of rare-earth elements (La, Pr, Nd, Sm, Eu, Gd with $0.94 \le R \le 1.04$ Å) in amounts up to 10-mol % was presented. Preliminary estimates showed that the observed effect is due to a decrease in the chemical bond strength ($\mu < 1.04$) in $BiFeO_3$ upon its modification and stabilization of the structure due to the shift of $BiFeO_3$ into the region of existence of the perovskite structure.

The presented results should be taken into account when developing devices using $BiFeO_3$/REE materials.

**Author Contributions:** K.P.A.—original draft preparation, writing—review and editing, investigation; V.P.S.—review and validation; A.V.T., A.A.P.—discussion of results and data curation; L.A.S.—X-ray investigation; S.I.D., A.G.R., D.D.R.—data curation; I.A.V.—resources, L.A.R.—conceptualization, S.V.H., I.A.P., S.-H.C., H.-Y.W.—discussion of results. All authors have read and agreed to the published version of the manuscript.

**Funding:** This work was financially supported by the Ministry of Science and Higher Education of the Russian Federation (State assignment in the field of scientific activity, Southern Federal University, 2020); Ministry of Science and Technology of the Republic of China (MOST108-2221-E-992-026-, MOST109-2221-E-992-091-).

**Conflicts of Interest:** The authors declare no conflict of interest.

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
