# Peer review of "Reasons for the High Electrical Conductivity of Bismuth Ferrite and Ways to Minimize It"

_applsci, doi:10.3390/app11031025_

Round 1

Reviewer 1 Report

The paper is very difficult to read, the figures are impossible to understand (scales are not visible, the labels missing, etc), some details about experiments are not relevant. For ex: the authors describe the method to determine the density but the densities of the samples are not presented in the paper. Also, why the authors describe the XRD with temperature? I didn't see the experimental data or the discussion of them. I didn't understand the scale in Fig.1 and for me it is not clear why the authors represent the ratio between real and imaginary part of permittivity and vacuum permittivity. In Fig.5 and 6 what represent the line? 

I consider that the paper needs important changes before reconsider for publication. 

Author Response

We would like to thank the reviewer for a careful reading of the manuscript and for a number of valuable comments and suggestions. These were carefully examined and implemented in the revised manuscript as indicated below.

The paper is very difficult to read, the figures are impossible to understand (scales are not visible, the labels missing, etc), some details about experiments are not relevant.

Response. Subsection Introduction was revised.

  1. A number of sentences have been simplified to make information easier to read.
  2. Added a number of references.

All figures in high resolution were uploaded to the current manuscript.

For ex:

Point 1. The authors describe the method to determine the density but the densities of the samples are not presented in the paper.

Response 1. Due to the large volume of the table containing data on densities of solid solutions Bi1-xREExFeO3 (REE - La, Pr, Nd, Sm, Eu, Gd, Tb, Dy, Ho, Er, Tm, Yb, Lu), a link to dataset was added to which provides all the necessary information. Text fragment was added page 3 line 102: Densities of the studied SS are presented in [12].

Point 2. Also, why the authors describe the XRD with temperature?  I didn't see the experimental data or the discussion of them.

Response 2. Text fragment was revised page 2 lines 92-95.

Point 3. I didn't understand the scale in Fig.1 and for me it is not clear why the authors represent the ratio between real and imaginary part of permittivity and vacuum permittivity.

Response 3. Usually, even if the authors give only the dielectric constant, it is assumed that the ratio was taken to the permittivity of the vacuum. For this reason, the ratio of the indicated values was shown on the ordinate axis.

Point 4. In Fig.5 and 6 what represent the line?

Response 4. Figure captions were revised:

Figure 5. Dependences of the electrical conductivity s of BiFeO3/REE – La, Pr, Eu, Gd, Tb, Ho, Dy, Er on the concentration of introduced x of REE (experimental points are marked with markers; solid lines serve as a guide to the eye).

Figure 6. Dependences of the values of specific electrical conductivity s, BiFeO3/REE on radii, R, REE (experimental points are marked with markers; solid lines serve as a guide to the eye).

Reviewer 2 Report

The paper presents electrical and  structural study on numerous bismuth ferrite samples, which were doped by rear earth elements. I have only three comments for the authors:

  1. A lot of samples have been prepared, but the preparation conditions are not given (only temperature intervals, heating duration "depending on the composition"). I strongly recommend to place a table with all materials list and their specific preparation conditions.
  2. Figures 5 and 6 only show electrical conductivities, so the figure captions should be corrected (there is no resistivity in the figures).
  3. Conclusions almost do not exist:

The authors write "...the main reasons for the formation of high electrical conductivity of bismuth ferrite, which impedes its practical use, have been revealed" - please summarize these reasons.

"...and ways to minimize it have been shown..." - please present all the ways discovered and summarize, why the substitutions work in the way, that the conductivity can be minimized.

Author Response

Reviewer #2:

We would like to thank the reviewer for a careful reading of the manuscript and for a number of valuable comments and suggestions. These were carefully examined and implemented in the revised manuscript as indicated below.

The paper presents electrical and structural study on numerous bismuth ferrite samples, which were doped by rear earth elements. I have only three comments for the authors:

Point 1. A lot of samples have been prepared, but the preparation conditions are not given (only temperature intervals, heating duration "depending on the composition"). I strongly recommend to place a table with all materials list and their specific preparation conditions.

Response 1. Due to the large volume of the table containing data on technological regulations for the preparation of solid solutions Bi1-xREExFeO3 (REE - La, Pr, Nd, Sm, Eu, Gd, Tb, Dy, Ho, Er, Tm, Yb, Lu), a link to dataset was added to which provides all the necessary information.

Text fragment was added page 2 line 8: Technological regimes for the preparing of SS Bi1-xREExFeO3 (REE – La, Pr, Nd, Sm, Eu, Gd, Tb, Dy, Ho, Er, Tm, Yb, Lu) are presented in [11].

Point 2. Figures 5 and 6 only show electrical conductivities, so the figure captions should be corrected (there is no resistivity in the figures).

Response 2. Figure captions were revised.

Figure 5. Dependences of the electrical conductivity s of BiFeO3/REE – La, Pr, Eu, Gd, Tb, Ho, Dy, Er on the concentration of introduced x of REE (experimental points are marked with markers; solid lines serve as a guide to the eye).

Figure 6. Dependences of the values of specific electrical conductivity s, BiFeO3/REE on radii, R, REE (experimental points are marked with markers; solid lines serve as a guide to the eye).

Point 3. Conclusions almost do not exist: The authors write "...the main reasons for the formation of high electrical conductivity of bismuth ferrite, which impedes its practical use, have been revealed" - please summarize these reasons.

"...and ways to minimize it have been shown..." - please present all the ways discovered and summarize, why the substitutions work in the way, that the conductivity can be minimized.

Response 3. Conclusion section was revised:

“Solid solutions of the composition Bi1-xREExFeO3 (rare-earth elements, lanthanides, Ln: La, Pr, Nd, Sm, Eu, Gd, Tb, Dy, Ho, Er, Tu, Yb, Lu; x = 0.0–0.5; ΔÑ… = 0.025, 0.05, 0.10) were prepared by two-stage solid-phase synthesis followed by sintering by conventional ceramic technology with varying annealing temperatures. Comprehensive studies of the structure and specific electrical conductivity of the obtained experimental samples have been carried out.

The study of the temperature dependences of ε′/ε0, ε′′/ε0 BiFeO3 and BiFeO3/REE on different frequencies of the ac field showed the presence of a low-temperature relaxation maximum ε′/ε0(T) at (400÷500) K, which is not associated with any of the known phase transitions in BiFeO3 and satisfies the Arrhenius law with Еа = 1.38 and 0.68 eV, ω0 = 2.8·1016 and 2.97·1014 rad/s, for ε′/ε0(T) and tanδ (T), respectively. The results obtained allowed us to conclude about the presence of Maskwell-Wagner relaxation, as well as to consider the physical mechanisms of macroevents developing in such heterogeneous media.

The possibility of a significant (by 2–3 orders of magnitude) decrease in the electrical conductivity of BiFeO3 by introducing into it large-size ions of rare-earth elements (La, Pr, Nd, Sm, Eu, Gd with 0.94≤ R ≤1.04 Å) in amounts up to 10 mol.% was presented. Preliminary estimates showed that the observed effect is due to a decrease in the chemical bond strength (μ <1.04) in BiFeO3 upon its modification and stabilization of the structure due to the shift of BiFeO3 into the region of existence of the perovskite structure.

The presented results should be taken into account when developing devices using BiFeO3/REE materials.”

Round 2

Reviewer 1 Report

The authors performed all the required modifications.